# Impact of Preparative Isolation of C-Glycosylflavones Derived from *Dianthus superbus* on In Vitro Glucose Metabolism

**DOI:** 10.3390/molecules29020339

**Published:** 2024-01-09

**Authors:** Zikai Lin, Xiaowei Zhou, Chen Yuan, Yan Fang, Haozheng Zhou, Zhenhua Wang, Jun Dang, Gang Li

**Affiliations:** 1Center for Mitochondria and Healthy Aging, College of Life Sciences, Yantai University, Yantai 264003, China; lzk1215741582@163.com (Z.L.); xiaoweiz1223@163.com (X.Z.); ytu_yc@163.com (C.Y.); fyy2008fx@163.com (Y.F.); bingfengtianyu@163.com (H.Z.); skywzh@ytu.edu.cn (Z.W.); 2Qinghai Provincial Key Laboratory of Tibetan Medicine Research, Key Laboratory of Tibetan Medicine Research, Chinese Academy of Sciences, Northwest Institute of Plateau Biology, Xining 810001, China

**Keywords:** C-glycosylflavones, *Dianthus superbus*, insulin resistance, two-dimensional liquid chromatography

## Abstract

*Dianthus superbus* L. has been extensively studied for its potential medicinal properties in traditional Chinese medicine and is often consumed as a tea by traditional folk. It has the potential to be exploited in the treatment of inflammation, immunological disorders, and diabetic nephropathy. Based on previous studies, this study continued the separation of another subfraction of *Dianthus superbus* and established reversed-phase/reversed-phase and reversed-phase/hydrophilic (RPLC) two-dimensional (2D) high-performance liquid chromatography (HPLC) modes, quickly separating two C-glycosylflavones, among which 2″-*O*-rhamnosyllutonarin was a new compound and isomer with 6‴-*O*-rhamnosyllutonarin. This is the first study to investigate the effects of 2″-*O*-rhamnosyllutonarin and 6‴-*O*-rhamnosyllutonarin on cellular glucose metabolism in vitro. First, molecular docking was used to examine the effects of 2″-*O*-rhamnosyllutonarin and 6″-*O*-rhamnosyllutonarin on AKT and AMPK; these two compounds exhibited relatively high activity. Following this, based on the HepG2 cell model of insulin resistance, it was proved that both of the 2″-*O*-rhamnosyllutonarin and 6‴-*O*-rhamnosyllutonarin demonstrated substantial efficacy in ameliorating insulin resistance and were found to be non-toxic. Simultaneously, it is expected that the methods developed in this study will provide a basis for future studies concerning the separation and pharmacological effects of C-glycosyl flavonoids.

## 1. Introduction

Diabetes has emerged as a rapidly rising global health issue in the 21st century [1,2]. Type II diabetes mellitus (T2DM), representing approximately 85 to 90% of the total population, has increased in prevalence from 4.5% in 1990 to 8.5% from 2016 to 2017 and is projected to rise further by 2030 [3]. Insulin resistance (IR) and dysfunction of pancreatic β-cells are factors that contribute to the progression of T2DM [4,5]. Patients with T2DM may secrete a small amount of insulin by themselves; therefore, hypoglycemic medicines are frequently used to induce insulin secretion from pancreatic β-cells and ameliorate the resistance of effector cells to insulin. It is crucial to identify natural substances that can regulate blood glucose without causing prominent side effects for T2DM treatment [6,7].

*Dianthus superbus* L. (*D. superbus*) is a medicinal drug derived from the dried aerial section of the perennial herb *Dianthus superbus* or *Dianthus chinensis*. Its primary production is predominantly concentrated in the regions of Shandong, Hebei, and Henan. *D. superbus* was first reported in the Shennong Classic of Materia Medica and was commonly used by ancient physicians in combination with different Chinese herbs to treat diseases such as heat stranguria, bloody stranguria, etc. [8,9]. Current studies have revealed that *D. superbus* mainly consists of anthocyanin, methyl salicylate, eugenol, vitamin A, saponins, and flavonoids that exhibit pharmacological properties including diuretic, antioxidant, hypotensive, and anti-neuritic effects. Most of the studies concerning *D. superbus* are confined to its chemical constituents, however, limited research explores its biological impacts and mechanisms.

Column chromatography and silica gel chromatography are traditional methods used for the extraction and purification of natural product compounds. However, both methods are afflicted with low resolution, weak separation operations, and insufficient reproducibility [10,11,12,13]. Thus, traditional approaches are ineffective for isolating high-purity compounds. Recent studies have highlighted high-speed counter-current chromatography as an innovative method for differentiating standard compounds from natural products [14,15,16]. Nevertheless, this approach continues to be hindered by insufficient separation resolution, the requirement for estimating the partition coefficient, and its capability to isolate only moderately polar compounds. Therefore, to achieve further advances in the therapeutic potential of *D. superbus*, it is crucial to find an alternative approach for the extraction and synthesis of high-purity compounds from this complex natural product.

The study of the modernization of Chinese medicine has led to a major focus on the separation of complex systems, which has emerged as a challenging and prominent area of study within the field of analytical chemistry. Reversed-phase liquid chromatography (RPLC) can solve a wide range of analytical applications but has little or no retention for hydrophilic and polar compounds. The investigation of *D. superbus* has been previously explored, in which a maltol glycoside was extracted and its anti-inflammatory efficacy was evaluated [17]. This study will continue to carry out the next step of research on *D. superbus*. Hydrophilic interaction liquid chromatography (HILIC) columns exhibit the opposite elution order as compared to RPLC. The substances that are challenging or non-retentive on RPLC columns exhibit strong retention on HILIC columns under normal experimental conditions. Therefore, the 2D chromatography technique of RPLC and HILIC coupling can complete the analysis of strong and weak polar substances and substantially enhance the efficiency of the analysis [18,19,20]. This study builds upon previous studies by focusing on the continued isolation of subfractions Fr41. The hypoglycemic activity of C-glycosylflavones derived from *D. superbus* was assessed in vitro using RPLC/RPLC and RPLC/HILIC modes.

The theoretical impact of C-glycosylflavones on AKT and AMPK was examined via molecular docking. To examine the impact of C-glycosylflavones derived from *D. superbus* on the regulation of insulin resistance in vitro, flow cytometry was performed to measure the uptake of 2-NBDG by cells, and Western blot was performed to monitor the levels of AMPK, AKT, and phosphorylated protein. This quick and efficient extraction method aids in the isolation and purification of *D. superbus*, as well as expanding its therapeutic activity by offering experimental and theoretical directions for the identification of lead compounds to address glucose metabolic disorders.

## 2. Results and Discussion

### 2.1. Separation of Fr41

The findings from the previous study indicated that the separation of *D. superbus* by medium-pressure preparative chromatography (Hanbon Science and Technology, Huai’an, China) yielded six subfractions, of which Fr41 yielded a final mass of 5.0 g (Figure 1A). Therefore, Fr41 was further separated based on the existing results. The analytical chromatogram for 1D analysis on a 7-X10 column (4.6 × 250 mm, 7 μm) is depicted in Figure 1(Ba). Following linear amplification, a 7-X10 preparation column (20 × 250 mm, 7 µm) was used for 1D separation. The preparation chromatogram is illustrated in Figure 1(Bb). These fractions were obtained and dried following five chromatographic cycles to extract Fr411 (2.1 g) and Fr412 (0.6 g).

A comparative analysis of Fr41 and Fr411 revealed that Fr411 was ineffectively separated on a 7-X10 column and Click XION column (Figure 2(Ab,Bb)). It is mentioned that practical analysis frequently uses RPLC/RPLC chromatography due to its appropriate mobile phase and high potential for each dimension peak [21,22]. By the experimental procedure mentioned in Section 2.2 (Figure 3A), a SunFire^®^ C18 column (analytical and preparative columns, Waters Corporation, Milford, CT, USA) was used to separate Fr411 (80 mg). The separated Fr4111 (28.62 mg) was obtained through the collection and drying of the fraction after five complete cycles of 1D separation.

A comparative analysis of Fr41 and Fr412 revealed that Fr412 was successfully separated on the Click XION column (Figure 2(Bc)) but was not able to be isolated on the 7-X10 column (Figure 2(Ac)). Consequently, the 2D RPLC/HILIC technique was implemented to separate the compounds with high purity [23,24,25]. As for Fr412, the analytical chromatogram on a Click XION analytical column with the above mentioned dimensions is shown in Figure 3(Ba), and the preparative chromatogram on a Click XION preparative column with respective dimensions is illustrated in Figure 3(Bb). Collecting the concentrated dried fraction after the completion of the 2D RPLC/HILIC experiments yielded fraction Fr4121 (49.92 mg). This further demonstrates that the selectivity and peak capacity of chromatographic separation can be enhanced via orthogonal purification.

### 2.2. Verification of the Purity and the Structural Identification of Fr4111 and Fr4121

The purity of Fr4111 and Fr4121 was assessed by using two columns: a SunFire^®^ C18 column and a Click XION column with the same dimensions: 4.6 × 250 mm, 5 μm. As illustrated in Figure 4A, the chromatograms for both Fr4111 and Fr421 demonstrated that their purity was ≥95%. By comparing the acquired ^1^H NMR (Bruker Avance NMR spectrometer, Ettlingen, Germany), MS (Agilent Instruments Co., Santa Clara, CA, USA), UV (Shimadzu UV2401PC instrument, Kyoto, Japan), IR (Bruker, Ettlingen, Germany), and ^13^C NMR (Bruker Avance NMR spectrometer) spectra with those of previous examinations, it was determined that Fr4111 corresponds to 2″-*O*-rhamnosyllutonarin and Fr4121 to 6‴-*O*-rhamnosyllutonarin. Figure 4B depicts the chemical structure of the investigated compound. Table 1 describes the 1H-NMR and 13C-NMR information of the studied compounds. 

The subsequent experiments were conducted using these two compounds as targets; however, 2″-*O*-rhamnosyllutonarin (Fr4111), (novel compound) is an isomer of 6‴-*O*-rhamnosyllutonarin (Fr4121).

### 2.3. Molecular Docking of 2″-O-rhamnosyllutonarin and 6‴-O-rhamnosyllutonarin

Regulation of metabolic disorders is dependent on AMPK and Akt signaling. The PI3K/Akt pathway serves as the primary mechanism by which insulin induces hypoglycemia, whereas the AMPK pathway mainly regulates glucose uptake in insulin-resistant states and is implicated in cellular differentiation, propagation, and growth [28,29,30].

Accordingly, potential interactions between AKT, AMPK, and 2″-*O*-rhamnosyllutonarin (Fr4111), 6‴-*O*-rhamnosyllutonarin (Fr4121), and the positive drug metformin were evaluated using molecular docking. In this analysis, considering AKT, Metformin was identified in a pocket containing abundant amino acids (GLU234, GLU278, THR291, ASP292), while 2″-*O*-rhamnosyllutonarin (Fr4111) was found in a pocket enclosed by abundant amino acids (LYS158, VAL164, ALA177, GLU228, ALA230, GLU234, MET281, ASP292) and 6‴-*O*-rhamnosyllutonarin (Fr4121) was found in a pocket enclosed by abundant amino acids (LYS158, PHE161, GLY162, VAL164, ALA177, THR211, MET227, GLU228, GLU234, GLU278, ASN279, MET281, THR291, ASP292) (Figure 5A). Binding energy values for Metformin, 2″-*O*-rhamnosyllutonarin (Fr4111), and 6‴-*O*-rhamnosyllutonarin (Fr4121) were −7.11 Kcal/mol, −5.41 Kcal/mol, and −5.65 Kcal/mol, respectively (Table 2). Metformin was identified within an AMPK compartment along with multiple amino acids (ARG83, THR85, ASP136, ER138), while 2″-*O*-rhamnosyllutonarin (Fr4111) was found in a pocket enclosed by multiple amino acids (ARG10, LYS12, ILE13, GLY14, ARG78, PRO79, SER84, PHE160) and 6‴-*O*-rhamnosyllutonarin (Fr4121) was found in a pocket enclosed by abundant amino acids (GLN50, VAL81, SER87, GLU139, PRO140, ILE152, ILE153, GLN154, ASP169) (Figure 5B). The values of binding energy for rofecoxib and tunicoside B were −5.90 Kcal/mol, −5.77 Kcal/mol, and −5.68 Kcal/mol, respectively. The findings of these analyses indicated that the minimum estimated binding energy for 2″-*O*-rhamnosyllutonarin and 6‴-*O*-rhamnosyllutonarin was comparable to that of Metformin, indicating favorable docking activity.

### 2.4. 2″-O-rhamnosyllutonarin and 6‴-O-rhamnosyllutonarin Can Enhance Glucose Uptake in IR-HepG2 Cells

Studies on IR are usually performed in insulin-induced insulin-resistant cells [31,32]. On the HepG2 cells, the IR model was efficiently developed in the current study. After 24 h of treatment with defined Fr4111 and Fr421 concentrations, the MTT assay did not detect any change in the HepG2 cell viability (Figure 6A). Insulin resistance, represented primarily by impaired glucose uptake, is the primary feature of T2DM. A major contributor to insulin resistance is obesity [33]. The fluorescent probe 2-NBDG is a fluorescent analog of 2-deoxyglucose, which can specifically bind intracellular glucose and has high sensitivity in detection. It has been widely used to explore cell metabolic functions. The glucose analog 2-NBDG was used to identify the glucose uptake capacity of IR-HepG2 cells. Generally, the fluorescence intensity of 2-NBDG in insulin-resistant cells or bodies is extremely weak, which is at the background value, indicating that cells or bodies have a weak ability to uptake glucose [34,35]. IR substantially reduces the cellular absorption of 2-NBDG, as illustrated in Figure 6B. A classical drug for the clinical treatment of T2DM, metformin (Met), increased insulin-stimulated glucose uptake in HepG2 cells by a substantial concentration of 1 mM. Additionally, glucose uptake by cells was enhanced in the treatment group treated with 2″-*O*-rhamnosyllutonarin (Fr4111) and 6‴-*O*-rhamnosyllutonarin (Fr4121) as opposed to the IR group.

### 2.5. P-AKT and P-AMPK Levels of 2″-O-rhamnosyllutonarin and 6‴-O-rhamnosyllutonarin

Western blot analysis exhibited levels of AKT, phosphorylated-AKT Ser473, AMPK, and phosphorylated-AMPK Thr172. This analysis provided details about the mechanisms by which 2″-*O*-rhamnosyllutonarin and 6″-*O*-rhamnosyllutonarin enhance glucose uptake in IR-HepG2 cells. Cell proliferation, growth, and metabolism are all significantly influenced by the PI3K/Akt pathway. This pathway is controlled by several mediators, including insulin-like growth factor (IGF) and epidermal growth factor (EGF), as well as the essential negative regulation of PTEN and other molecular pairs to maintain cell and life homeostasis. The PI3K/Akt signaling pathway represents the predominant downstream molecular pathway associated with insulin. An increase in the phosphorylation level of AKT2 is commonly attributed to the metabolic function of insulin. Specifically, when the body or cells enter the IR state, the phosphorylation level of AKT decreases [36,37,38]. Insulin stimulates AKT2, which is widely regarded as performing its most vital function.

An essential protein kinase, AMPK is crucial for the regulation of energy and metabolism. Important functions of activated AMPK include regulating gene transcription, increasing fatty acid oxidation, and enhancing insulin sensitivity and glucose uptake by skeletal muscle. This AMPK may serve as an innovative pharmacological therapeutic target for obesity, insulin resistance, and T2DM due to its capacity to regulate glucose and fatty acid metabolism [39,40,41].

Data presented in Figure 6C,D represent the effects of 2″-*O*-rhamnosyllutonarin and 6‴-*O*-rhamnosyllutonarin on the AKT and AMPK stimulation. The findings reveal that normal HepG2 cells did not exhibit activation of AKT and AMPK, resulting in a relatively low level of protein expression. In contrast to the control group, the phosphorylated protein expression of AKT was remarkably reduced in the IR-HepG2 group when compared to the positive control (Met, 1 mM). This finding confirms that the proposed model has been effectively developed. Moreover, both Fr4111 and Fr4121 can substantially stimulate AKT and AMPK, which also verify the experimental results of molecular docking.

## 3. Materials and Methods

### 3.1. Sample Separation and Purity Assay

Based on the previous outcomes, the separation of *D. superbus* by medium-pressure preparative chromatography yielded six subfractions, of which Fr41 yielded a final mass of 5.0 g. The ID separation was then carried out using a 7-X10 preparative column (analytical and preparative, 20 × 250 mm, 7 μm) (Acchrom Tech, Beijing, China). The mobile phase was chromatography-grade methanol (CH_3_OH), acetonitrile (ACN) (Yunnan Xinlanjing Chemical Industry, Yuxi, Yunnan, China) (B) and 0.1% of formic acid in H_2_O (*v*/*v*). The chromatogram was acquired at 210 nm after the sample was isocratically eluted at a flow rate of 19 mL/min in 6% B for 90 min. The fraction (Fr41) was divided into Fr411 (0.6 g) and Fr412 (2.1 g) subfractions.

For comparison, Fr41, Fr411, and Fr412 were analyzed via 7-X10 and Click XION analytical columns, respectively. An isocratic elution phase of 0 to 70 min on a 7-X10 column (4.6 × 250 mm, 7 μm) containing 6% CH_3_OH; a Click XION column (4.6 × 250 mm, 5 μm) exhibiting a linear gradient of 0 to 60 min; and 85 to 80% ACN. At 210 nm, chromatograms were recorded.

Approximately 80 mg of Fr411 was measured to facilitate the next preparation steps. For the 2D separation, a SunFire^®^ column (20 × 250 mm, 5 μm) was selected. The mobile phase and flow rate were the same as in the previous step. The elution conditions were 0–80 min with 20% B isocratic elution. At 210 nm, the chromatogram was acquired, and the target fraction was identified, yielding 28.62 mg of Fr4111.

In the 2D separation step, a Click XION column (20 × 250 mm, 5 μm) was used for Fr412 analysis. Isocratic elution with 90% ACN at a flow rate of 19 mL/min in 90 min (5% *v*/*v* trifluoroacetic acid in H_2_O was another mobile phase) was performed. The target fraction Fr4121 was observed at 210 nm. Following collection and concentration, the final mass was 49.92 mg.

Purity verification of Fr4111 and Fr4121 was performed on SunFire^®^ C18 and Click XION columns (4.6 × 250 mm, 5 µm).

The purity of Fr4111 and Fr4121 was verified via SunFire^®^ C18 and Click XION columns (4.6 × 250 mm, 5 µm). The elution condition on the SunFire^®^ C18 column was a gradient from 20% to 80% CH_3_OH over 0 to 90 min. The elution requirement on the Click XION column is 0 to 90 min with a gradient of ACN from 90 to 60%. The elution flow rates were set at a constant value of 1 mL/min. Chromatograms were acquired at 210 nm.

### 3.2. Molecular Docking

Molecular docking is a commonly used drug screening technique that relies on the 3D conformation of target proteins. In this study, the protein IDs corresponding to AKT (6CCY PDB ID) and AMPK (5KQ5 PDB ID) were found in the RCSB PDB database (https://www.rcsb.org/, accessed on 6 May 2023), and the protein IDs were entered from the PDB protein database to examine the 3D structures of the proteins as the macromolecular receptors for molecular docking. The proteins’ small molecule ligands and water molecules were removed via PyMOL 2.5 software. Following hydrogenation, the ligands and receptors were docked through AutoDock 4.2.6 software. Lastly, only those molecular docking outcomes that possessed the least amount of free energy of each docking pair were retained [42,43].

### 3.3. Cell Culture

Cells (human hepatocellular carcinoma HepG2 cells) were collected from the cell bank of the Shanghai Institute of Biochemistry and Cell Culture (Shanghai, China) and allowed to grow in 10% fetal bovine serum (FBS) (Gibco, Billings, MT, USA) containing Dulbecco’s Modified Eagle’s Medium (DMEM) (Gibco, USA) under sterile or standard culture conditions.

### 3.4. MTT Assay

Cell viability was monitored via MTT assay. After 24 h of HepG2 cell (5 × 10^3^ cells/well) seeding into cultured 96-well plates, they were allowed to grow and treated with defined concentrations (0, 5, 10, 20, 50 μM) of compounds for 24 h. Treated cells were exposed to MTT dye (5 mg/mL, 10 μL/well) and kept in the dark at 37 °C for 4 h. Subsequently, DMSO (100 μL/well) was used to replace the media. The final absorbance (OD) was noted at 490 nm by a microplate reader (Molecular Devices, San Jose, CA, USA).

### 3.5. 2-NBDG Uptake Assay

Following the modeling procedure, treatments of Fr4111 (10 µM), Fr4121 (10 µM), and Met (1 mM) were added. Next, the medium was replaced with DMEM without glucose. Following a 2 h incubation, 100 μM 2-NBDG was introduced into the medium and maintained for a further 30 min. After PBS washes (thrice), the cells were detached and collected in the dark. The FACS AriaTM flow cytometer (Becton Dickinson, Franklin Lakes, NJ, USA) was used to quantify the fluorescence (485/20 and 540/20 nm excitation/emission respectively). The average fluorescence signal of 10,000 cells was measured and the data were displayed.

### 3.6. Western Blot Analysis

Following the cell treatment, cells were rinsed twice with chilled PBS, and total proteins were isolated via lysis buffer (200 μL). The resulting lysates were centrifuged at 14,000× *g*, 4 °C, for 15 min. The concentration of protein in the supernatant was assessed via the BCA (Beyotime, China) assay. Proteins were transferred to a PVDF membrane following separation by SDS-PAGE. Following 1 h of blocking at 25 °C with 5% skim milk, membranes were kept at 4 °C overnight with antibodies targeting AMPK, p-AKT, AKT, and p-AMPK (Cell Signaling, Shanghai, China) (1:1000). The membrane was exposed to HRP goat anti-rabbit antibody for 2 h at 25 °C. The protein assay was conducted as per the guidelines outlined in the enhanced chemiluminescent (ECL) assay (Beyotime, Shanghai, China) kit. The bands of protein with β-actin (control) were detected via a 5200 Multi-Luminescent Image Analyzer (Tanon Science and Technology, Shanghai, China).

### 3.7. Statistical Analysis

All presented data exhibited the means ± SD of replicated experiments. Statistical comparisons were noted via one-way ANOVA or Student’s *t*-test with the SPSS 22.0 software (SPSS, Chicago, IL, USA). A threshold level of the *p*-value was specified as *p* ≤ 0.05.

## 4. Conclusions

This study identified a new subfraction of *D. superbus*, Fr41. Two C-glycosylflavones, 2″-*O*-rhamnosyllutonarin (Fr4111) and 6″-*O*-rhamnosyllutonarin (Fr4121) were extracted from *D. superbus* via 2D HPLC in RPLC/RPLC and RPLC/HILIC, and 2″-*O*-rhamnosyllutonarin was identified as a novel compound isomeric of 6‴-*O*-rhamnosyllutonarin. The results of molecular docking showed that both compounds exhibited good binding energy with AKT and AMPK. Following IR-HepG2 cell model development, both extracted compounds were tested for insulin resistance improvement. They enhanced 2-NBDG uptake, AKT, and AMPK pathways, and reduced IR symptoms. It was the first preliminary study that reported the 2″-*O*-rhamnosyllutonarin and 6‴-*O*-rhamnosyllutonarin isolation from *D. superbus*. Further research is required to elucidate their mechanistic role and discover the precise glucose metabolism of these compounds. Collectively, it is expected that the methods developed in this study on C-glycosylflavones derived from *D. superbus* will provide a future direction for the separation and pharmacological effects of C-glycosylflavones.

## Figures and Tables

**Figure 1 molecules-29-00339-f001:**
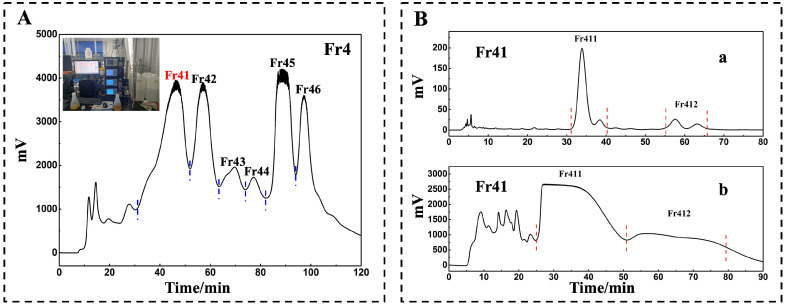
A fraction of *D. superbus*-derived Fr41 was separated by Fr4 pretreatment chromatograms and 1D RPLC. A Spherical C18 preparative column was used to generate Fr4 (**A**). The following settings were applied to a 7-X10 analytical column (4.6 × 250 mm, 7 μm) to analyze Fr41; mobile phase A: 0.1% of formic acid in chromatographic grade H_2_O (*v*/*v*), B: CH_3_OH, gradient: 0 to 90 min, 6% B, a chromatogram at 210 nm was noted (**Ba**). A 7-X10 preparative column with defined dimensions was used to achieve 1D separation of Fr41. The column was designed as follows: A: 0.1% of formic acid in chromatographic grade H_2_O (*v*/*v*), B: CH_3_OH, gradient: 0 to 90 min 6% B, a chromatogram was noted at 210 nm (**Bb**).

**Figure 2 molecules-29-00339-f002:**
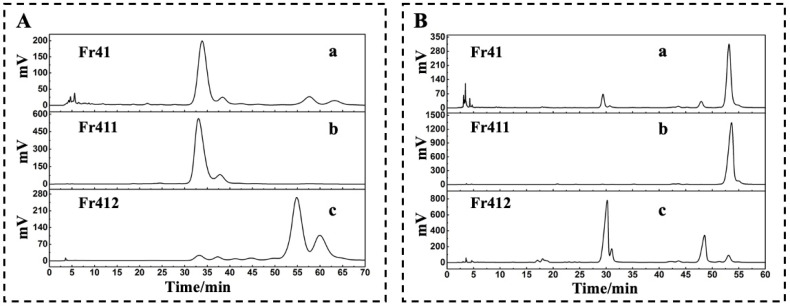
Comparative chromatographic profiles of Fr41, Fr411, Fr412. Fr41, Fr411, and Fr412 were examined via a 7-X10 analytical column with respective dimensions. The column was adjusted as follows: mobile phase A: 0.1% of formic acid in chromatographic grade H_2_O (*v*/*v*), B: CH_3_OH; gradient: 0 to 70 min, 6% B (**A**). To evaluate Fr41, Fr411, and Fr412, a Click XION analytical column with defined dimensions was used with the following respective setups: mobile phase A: 0.1% of formic acid in chromatographic grade H_2_O (*v*/*v*), B: ACN, gradient: 0 to 60 min, 85 to 80% B, a chromatogram at 210 nm was noted (**B**).

**Figure 3 molecules-29-00339-f003:**
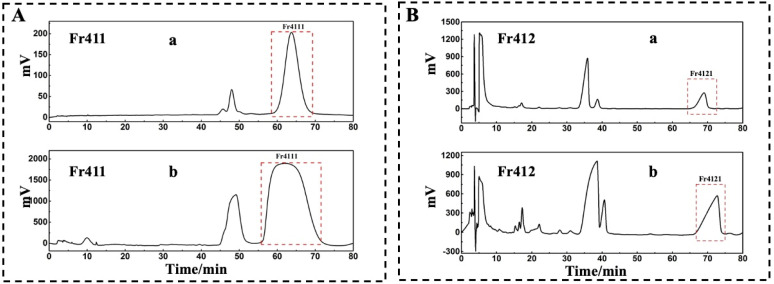
In RPLC/HILIC and 2D RPLC/RPLC modes, Fr411 and Fr412 were separated. Fr411 was observed with a SunFire^®^ C18 analytical column (4.6 × 250 mm, 7 μm) with the following respective parameters: mobile phase A: 0.1% of formic acid in chromatographic grade H_2_O (*v*/*v*), B: CH_3_OH, gradient: 0 to 80 min, 20% B, a chromatogram was noted at 210 nm (**Aa**). For 2D separation, a SunFire^®^ C18 preparative column (20 × 250 mm, 5 μm) was used with these subsequent setups: mobile phase A: 0.1% of formic acid in chromatographic grade H_2_O (*v*/*v*), B: CH_3_OH, gradient: 0 to 80 min, 20% B, a chromatogram was noted at 210 nm (**Ab**). The Click XION analytical column (4.6 × 250 mm, 7 μm) was used to examine Fr412. The column was set up as follows: mobile phase A: chromatographic grade H_2_O containing 5% trifluoroacetic acid *v*/*v*, B: ACN, gradient: 0 to 80 min, 90% B, a chromatogram was noted at 210 nm (**Ba**). For 2D separation, a Click XION preparative column with defined dimensions was used with the respective arrangements: mobile phase A: chromatographic grade H_2_O containing 5% trifluoroacetic acid *v*/*v*, B: ACN, gradient: 0 to 80 min, 90% B, a chromatogram was noted at 210 nm (**Bb**).

**Figure 4 molecules-29-00339-f004:**
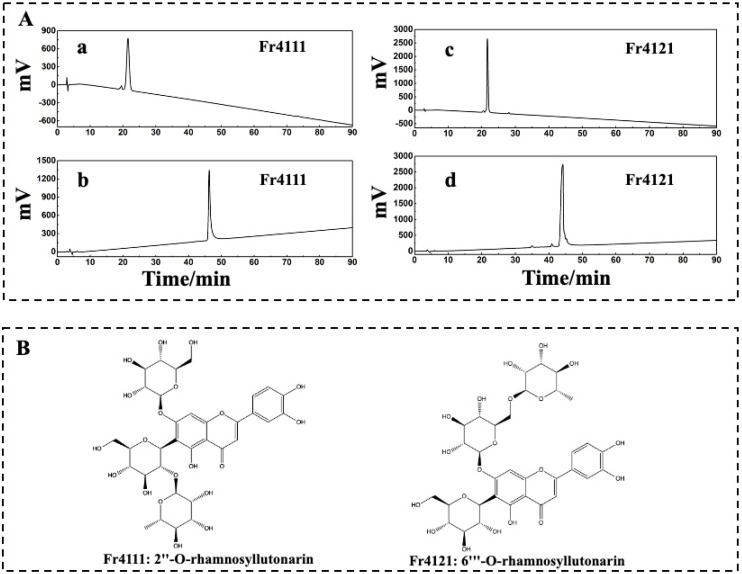
Analysis of the chemical structures and purity of Fr4111 and Fr4121. A SunFire^®^ C18 column with defined dimensions was used to evaluate Fr4111 and Fr4121 at the subsequent parameters: mobile phase A: 0.1% of formic acid in chromatographic grade H_2_O (*v*/*v*), B: CH_3_OH; gradient: 0 to 90 min, 20 to 80% B (**Aa**,**Ac**), and with a Click XION column, with the following respective setup: mobile phase A: 0.1% of formic acid in chromatographic grade H_2_O (*v*/*v*), B: ACN; gradient: 0 to 90 min, 90 to 60% B (**Ab**,**Ad**). The chemical structure of the Fr4111 and Fr4121 compounds, extracted from *D. superbus* (**B**).

**Figure 5 molecules-29-00339-f005:**
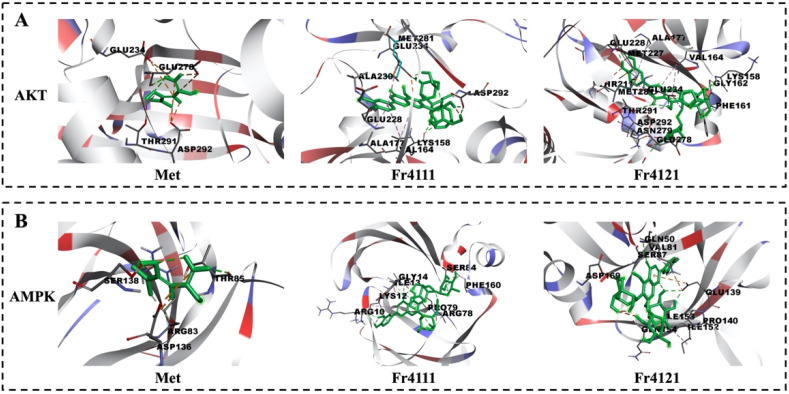
Analysis of the AKT’s interaction with 2″-*O*-rhamnosyllutonarin (Fr4111), 6‴-*O*-rhamnosyllutonarin (Fr4121), and Metformin via molecular docking (**A**). Molecular docking analysis of AMPK binding to 2″-*O*-rhamnosyllutonarin (Fr4111), 6‴-*O*-rhamnosyllutonarin (Fr4121), and Metformin (**B**). Metformin, 2″-*O*-rhamnosyllutonarin (Fr4111), and 6‴-*O*-rhamnosyllutonarin (Fr4121) structures are shown in green.

**Figure 6 molecules-29-00339-f006:**
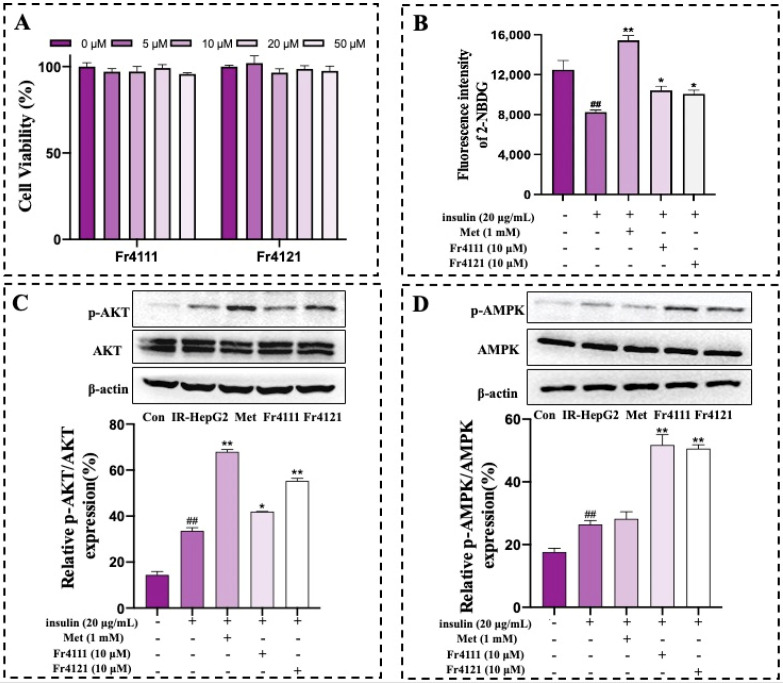
The effects of various concentrations of 2″-*O*-rhamnosyllutonarin (Fr4111) and 6‴-*O*-rhamnosyllutonarin (Fr4121) on the HepG2 cells were evaluated via the MTT assay (**A**). The quantification of glucose uptake by IR-HepG2 cells was assessed via flow cytometry analysis using 2-NBDG as a fluorescent probe (**B**). The *p*-value, ^##^
*p* ≤ 0.01 in contrast to control cells, * *p* ≤ 0.05, and ** *p* ≤ 0.01 compared to IR-HepG2 cells. At least three independent experiments were conducted to evaluate duplicate samples. Western blot analysis quantified the phosphorylation levels of AMPK and AKT (**C**,**D**). The *p*-value, ^##^
*p* ≤ 0.01 relative to normal cells, * *p* ≤ 0.05, and ** *p* ≤ 0.01 relative to IR-HepG2 cells.

**Table 1 molecules-29-00339-t001:** Information of the isolate from *D. superbus*.

Compound	Name	Structure	ESI^−^	^1^H-NMR (600 MHz, DMSO-*d*_6_)	^13^C-NMR (151 MHz, DMSO-*d*_6_)
**Fr4111**	2″-*O*-rhamnosyllutonarin	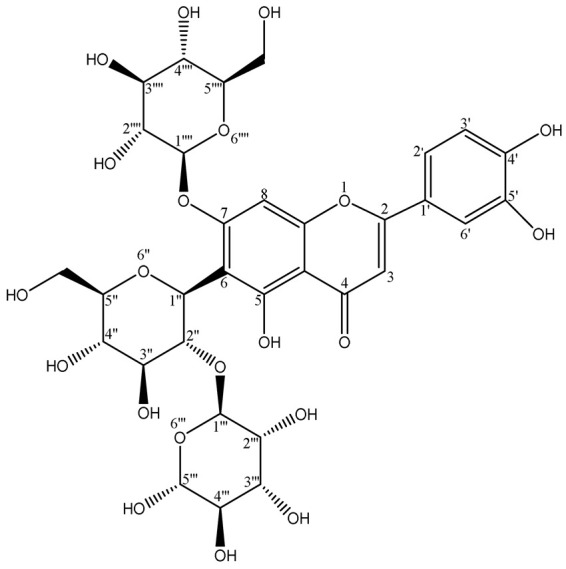	755.44	13.59 (1H, s, 5-OH), 7.44 (1H, dd, *J* = 8.2, 2.0 Hz, H-6′), 7.42 (1H, brs, H-2′), 6.90 (1H, d, *J* = 8.2 Hz, H-5′), 6.87 (1H, s, H-8), 6.75 (1H, s, H-3), 4.97 (1H, d, *J* = 7.9 Hz, H-1⁗), 4.95 (1H, brs, H-1‴), 4.71 (1H, d, *J* = 9.9 Hz, H-1″), 4.13 (1H, dd, *J* = 9.9, 8.5 Hz, H-2″), 3.61 (1H, brs, H-2‴), 3.58, 3.64 (each 1H, m, H-6″), 3.56, 3.78 (each 1H, m, H-6⁗), 3.50 (1H, m, H-5⁗), 3.37 (1H, m, H-2⁗), 3.36 (1H, m, H-3″), 3.32 (1H, m, H-3⁗), 3.28 (1H, m, H-4″), 3.24 (1H, m, H-4⁗), 3.18 (1H, m, H-5″), 3.15 (1H, dd, *J* = 9.3, 3.0, H-3‴), 2.88 (1H, dd, *J* = 9.3, 9.3, H-4‴), 2.32 (1H, dq, *J* = 9.3, 6.2, H-5‴), 0.59 (3H, d, *J* = 6.2, H-6‴) [26].	181.9 (C-3), 164.3 (C-1), 162.7 (C-6), 159.3 (C-4), 156.6 (C-8), 150.2 (C-4′), 145.9 (C-3′), 121.1 (C-1′), 119.2 (C-6′), 116.0 (C-5′), 113.4 (C-2′), 110.5 (C-5), 105.0 (C-9), 103.1 (C-2), 102.0 (C-1⁗), 100.5 (C-1‴), 94.3 (C-7), 81.1 (C-5″), 78.9 (C-3″), 77.3 (C-5⁗), 76.7 (C-2″), 75.7 (C-3⁗), 73.8 (C-2⁗), 71.6 (C-4″), 71.2 (C-1″), 70.5 (C-2‴), 70.2 (C-3‴), 69.7 (C-4″,4⁗), 68.3 (C-5‴), 60.9 (C-6⁗), 60.3 (C-6″), 17.6 (C-6‴) [26].
**Fr4121**	6‴-*O*-rhamnosyllutonarin	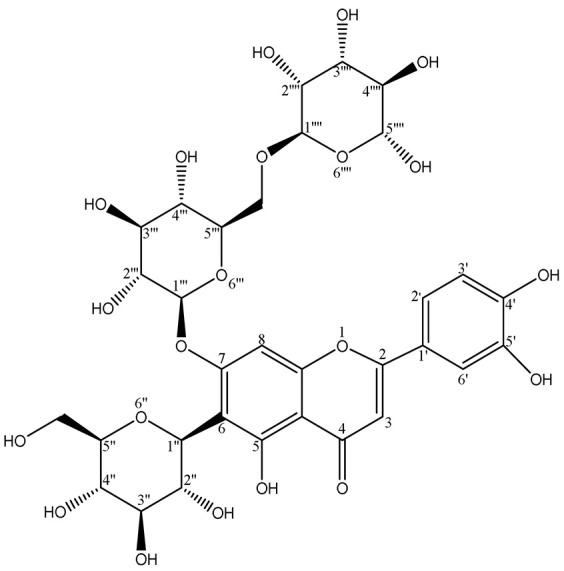	755.42	13.61 (1H, s, 5-OH), 7.45 (1H, dd, *J* = 8.4, 2.0 Hz, H-6′), 7.41 (1H, d, H-2′), 6.95 (1H, d, *J* = 8.4 Hz, H-5′), 6.77 (1H, s, H-3), 6.75 (1H, s, H-8), 4.99 (1H, d, *J* = 7.1 Hz, H-1‴), 4.65 (1H, d, *J* = 9.8 Hz, H-1″), 4.63 (1H, brs, H-1⁗), 3.94 (1H, dd, *J* = 9.8, 8.7 Hz, H-2″), 3.93 (1H, brd, *J* = 12.0, H-6‴), 3.73 (1H, dd, *J* = 3.3, 1.4 Hz, H-2⁗), 3.68 (1H, m, H-5‴), 3.55 (1H, dd, *J* = 9.4, 3.3 Hz, H-3⁗), 3.54–3.65 (2H, m), 3.51 (1H, m, H-6‴), 3.46 (1H, dq, *J* = 9.3, 6.2 Hz, H-5⁗), 3.34 (1H, m, H-2‴), 3.33 (1H, m, H-3‴), 3.28 (1H, m, H-4″), 3.22 (1H, m, H-3″), 3.20 (1H, m, H-4⁗), 3.19 (1H, m, H-4‴), 3.17 (1H, m, H-5″), 1.13 (3H, d, *J* = 6.2, H-6⁗) [27].	182.0 (C-4), 164.7 (C-2), 162.5 (C-7), 159.5 (C-5), 156.6 (C-9), 150.0 (C-4′), 145.7 (C-3′), 121.2 (C-1′), 119.4 (C-6′), 116.3 (C-5′), 113.5 (C-2′), 110.5 (C-6), 105.0 (C-10), 103.3 (C-3), 101.3 (C-1‴), 100.6 (C-1⁗), 93.5 (C-8), 81.2 (C-5″), 78.9 (C-3″), 75.7 (C-5‴), 75.5 (C-3‴), 73.8 (C-2‴), 72.8 (C-1″), 72.0 (C-4⁗), 70.9 (C-3⁗), 70.8 (C-2″), 70.4 (C-2⁗), 69.6 (C-4″,4‴), 68.4 (C-5⁗), 66.2 (C-6‴), 60.3 (C-6″), 17.9 (C-6⁗) [27].

**Table 2 molecules-29-00339-t002:** The binding energy of 2″-*O*-rhamnosyllutonarin (Fr4111), 6‴-*O*-rhamnosyllutonarin (Fr4121), and Metformin in AKT and AMPK.

Type	Compounds	RMSD	Binding Energy (Kcal/mol)
AKT	Metformin	33.50	−7.11
	2″-*O*-rhamnosyllutonarin	34.15	−5.41
	6‴-*O*-rhamnosyllutonarin	32.63	−5.65
AMPK	Metformin	99.38	−5.90
	2″-*O*-rhamnosyllutonarin	89.32	−5.77
	6‴-*O*-rhamnosyllutonarin	85.10	−5.68

## Data Availability

The data presented in this study are available in article and Appendix A.

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
