# Peer review of "Impact of Preparative Isolation of C-Glycosylflavones Derived from Dianthus superbus on In Vitro Glucose Metabolism"

_molecules, 2024, doi:10.3390/molecules29020339_

Round 1

Reviewer 1 Report

Comments and Suggestions for Authors

The manuscript describes the isolation, structure, and biological activity of C-glycosylflavones from plant Dianthus superbus. All looks fine. I would recommend to present NMR data in the tables instead of linear text.

Author Response

Thank you for your pertinent and detailed comments. All of these comments are valuable and helpful for us to revise and improve the paper, and they are also important guidance for our next research. According to your comments and suggestions, we have revised this manuscript carefully. We hope that this revision can make our paper recognized by you and readers. All the revisions in the text are marked in yellow background. The following are the answers to your questions.

Point 1: The manuscript describes the isolation, structure, and biological activity of C-glycosylflavones from plant Dianthus superbus. All looks fine. I would recommend to present NMR data in the tables instead of linear text.

Response 1: Thank you very much for your pertinent suggestions, and we have revised the manuscript according to your suggestions by organizing the NMR data into a tabular form for presentation. In addition, we have further embellished the language throughout the manuscript without changing the original intent to further enhance the readability of the manuscript and make it easier for readers to understand. Specific changes have been highlighted in yellow in the revised manuscript. Finally, thank you again for your suggestions.

Reviewer 2 Report

Comments and Suggestions for Authors

The abstract and introduction warrant a comprehensive revision, as instances of unclear ideas and poor language proficiency are evident throughout. A thorough rewriting of these two sections is essential to ensure that the narrative not only resonates with clarity but also reflects a  command of the English language

The supplementary materials do not require any corrections

The work done, methodologies and the result section are fine, please refer to the comments in the attached document for further information

Comments on the Quality of English Language

The quality of language needs to be improved especially for abstract and introduction.

Author Response

Dear Reviewer,

Thank you for your pertinent and detailed comments. All these comments are valuable and helpful for us to revise and improve the paper, and they are also important guidance for our next research. According to your comments and suggestions, we have revised this manuscript carefully. We hope that this revision can make our paper recognized by you. All the revisions in the text are marked with yellow background. Detailed responses to your questions and other documents will be listed in the attachment.

Yours sincerely,

Zikai Lin

Zikai Lin  

Reviewer 3 Report

Comments and Suggestions for Authors

The authors, Zikai Lin at al discussed the “Preparative Isolation of C-glycosylflavones from Dianthus 2 superbus and Their Effects on Glucose Metabolism In vitroDianthus superbus is a traditional Chinese medicine and can also be used as a tea in the
folk. It can be applied as a treatment for inflammation, immunological disorders and improve dia-
betic nephropathy. On the basis of previous studies, this study continued to separate another sub-
fraction of Dianthus superbus and established reversed-phase/reversed-phase and reversed-
phase/hydrophilic two-dimensional high-performance liquid chromatography mode, quickly sepa-
rated two C-glycosylflavones, among which 2’’-O-rhamnosyllutonarin was a new compound and
was isomer with 6’’’-O-rhamnosyllutonarin. Then the insulin
resistance model was established by treating HepG2 cells with insulin at a concentration of 20
μg/mL for 36 h. Based on the successful establishment of the IR-HepG2 model, it was investigated
whether the 2’’-O-rhamnosyllutonarin and 6’’’-O-rhamnosyllutonarin could improve the insulin re-
sistance condition. It was found that both 2’’-O-rhamnosyllutonarin and 6’’’-O-rhamnosyllutonarin
were non-toxic and exhibit significant activity in improving insulin resistance. At the same time, the
method established in this study is expected to provide a theoretical basis for the preparation, sep-
aration and pharmacological activity of C-glycosyl flavonoids in the future.  

In this manuscript, some grammatical and scientific mistakes are observed. So, I suggest this manuscript for publishing in the journal of “Moleculeswith following instructions:

1)    In scientific language all the physical units and symbols must be italic author must correct it in whole manuscript where it is applicable.

2)    Volume number are missing in some references author should consider it in order to maintain uniformity.

3)    Conclusion is too long author should discuss it concisely.

1)    Author should discuss some considerable numerical findings in the conclusion and just focusing on your outcomes.

2)    Headings and sub-headings should be bold in the manuscript, author should consider it.

3)    Figure 4 structures are blur, author should re-draw it.

4)    Figure 6 labeling is not readable author should consider it.

5)    In scientific world et al must be italic author must acknowledge it and modify manuscript according to it.

6)    What is the primary objectives and aims of the study, author should discuss in the introduction?

7)    All the abbreviations should be defined in its first appearance.

8)    In reference 42, the year of publishing is not bold author should consider it.

9)    All the references should be rechecked to maintain the uniformity.

10) Language of the manuscript should be thoroughly checked as many grammatical errors are found in the manuscript.

11) Some recent reference related to this work should be cited.

12) All the headings and sub-headings should be bold, author should consider it.

13) Irrelevant spaces are observed in the whole manuscript author should consider it.

14) In figure 4 (B) structure labeling are observed in red color writing, author should recheck it.

15) What were the key findings or insights obtained from the molecular docking experiments?   

16) In “Molecular docking of 2’’-O-rhamnosyllutonarin and 6’’’-O-rhamnosyllutonarin” section, author should discuss some intro about docking with the help of following articles and cite them,

Journal of Molecular Structure1245, 131063, https://doi.org/10.1016/j.molstruc.2021.131063

Journal of Molecular Structure1175, 889-899, https://doi.org/10.1016/j.molstruc.2018.08.048

Comments on the Quality of English Language

its ok

Author Response

Dear reviewer,

Thank you for your pertinent and detailed comments. All these comments are valuable and helpful for us to revise and improve the paper, and they are also important guidance for our next research. According to your comments and suggestions, we have revised this manuscript carefully. We hope that this revision can make our paper recognized by you and readers. All the revisions in the text are marked with yellow background. Detailed responses to your questions and other documents will be listed in the attachment.

Yours sincerely,

Zikai Lin

Reviewer 4 Report

Comments and Suggestions for Authors

Two chimicals were seperated and identified, which has potential effects on glucose metabolism. There are some places can be improved

1) The full name should be used for the first occurrence of English abbreviations. such as NBDC, ***

2) All the references should be number sequence but not mixture of number and name year. lines 145-146: Wang et al., 2020, Wang  et al., 2017

3) Someplace of figures should be revised, such as c and d were missed in Figure 3B.

4) Lines 115- : I think the result should be presented and then discussion.

5) All the methods should have reference(s).

6) "3.1 Apparatus and chemicals" delete and then intuduce them after they appeared in text.

7) The conclusion is too complex. Simplified to identify two substances and their effects, what is their potential value.

Comments on the Quality of English Language

Line 28-29: I think it is not a completed sentence.

Line 30: I think it should be result but no the method.

lines 257-260: they can be combined in one table.

Author Response

(The authors gave the same response as above.)

Round 2

Reviewer 3 Report

Comments and Suggestions for Authors

Authors fulfilled all the comments and the article is improved so it may be published in current form